# Silk Fibroin/ZnO Coated TiO_2_ Nanotubes for Improved Antimicrobial Effect of Ti Dental Implants

**DOI:** 10.3390/ma16175855

**Published:** 2023-08-26

**Authors:** Angela Gabriela Păun, Cristina Dumitriu, Camelia Ungureanu, Simona Popescu

**Affiliations:** Faculty of Chemical Engineering and Biotechnologies, National University of Science and Technology Politehnica Bucharest, Gheorghe Polizu 1-7 Street, 011061 Bucharest, Romania; angela.olaru@upb.ro (A.G.P.); dumitriu.cristina.o@gmail.com (C.D.); ungureanucamelia@gmail.com (C.U.)

**Keywords:** TiO_2_ nanotubes, ZnO nanoparticles, silk fibroin, polydopamine, antibacterial effect

## Abstract

The aim of the present research is to develop a novel hybrid coating for a Ti dental implant that combines nature-inspired biomimetic polymers and TiO_2_ nanostructures with an entrapped ZnO antimicrobial agent. ZnO was used in other studies to cover the surface of Ti or Ti–Zr to reduce the need of clinical antibiotics, prevent the onset of peri-implantitis, and increase the success rate of oral clinical implantation. We developed an original coating that represents a promising approach in clinical dentistry. The titanium surface was first anodized to obtain TiO_2_ nanotubes (NT). Subsequently, on the NT surface, silk fibroin isolated from Bombyx mori cocoons was deposited as nanofibers using the electrospun technique. For an improved antibacterial effect, ZnO nanoparticles were incorporated in this biopolymer using three different methods. The surface properties of the newly created coatings were assessed to establish how they are influenced by the most important features: morphology, wettability, topography. The evaluation of stability by electrochemical methods in simulated physiological solutions was discussed more in detail, considering that it could bring necessary information related to the behavior of the implant material. All samples had improved roughness and hydrophilicity, as well as corrosion stability (with protection efficiency over 80%). The antibacterial test shows that the functional hybrid coating has good antibacterial activity because it can inhibit the proliferation of *Staphylococcus aureus* up to 53% and *Enterococcus faecalis* up to 55%. All Ti samples with the modified surface have proven superior properties compared with unmodified TiNT, which proved that they have the potential to be used as implant material in dentistry.

## 1. Introduction

The most common problem in dentistry is the loss of dental and/or periodontal structures, which is usually caused by dental caries, dental abnormalities, traumatic injuries, periodontal disease, and systemic disease [1]. Teeth have a limited capacity for self-regeneration; consequently, it is necessary to replace the damaged bone part with a graft that has high biocompatibility, osteoinductivity, osteoconductivity, osteointegrativity, and suitable mechanical properties [2,3]. Replacing missing teeth with a dental implant is an operation with a high success rate, good predictability, versatility, and fewer complications [4,5].

Metallic materials are used in various dental applications, such as filling cavities, building bridges and wires, or in dental implants [6,7,8,9]. Among the metallic materials, titanium is the most frequently employed in dental applications [10] because of its non-toxicity, hardness, light weight, excellent resistance, biocompatibility, and elastic modulus similar to human hard tissue [6,10,11,12,13]. Although titanium forms an inert layer of TiO_2_, when it contacts the in vivo microenvironment, it lacks the capacity to resist corrosion and bind to tissue. Moreover, it needs antibacterial protection. There have been progressive implant failures associated with peri-implant illnesses (peri-mucositis and peri-implantitis) [14]. Because systemic dosages may not be effective in overcoming the constraints of traditional therapy, the local injection of medications to the bone is also acknowledged as a possible alternative [15]. Therefore, surface modification for titanium is frequently necessary to enhance its bio-performance as an implant [11,12,16,17,18]. Choosing biomimetic dental implants with antibacterial properties (in such a way to prevent multidrug resistance) in those with a history of periodontitis may help avoid peri-implantitis, avoiding recurring visits to the doctor for debridement and significantly decreasing expenses and patient suffering [14].

The nanotubular titanium dioxide (NT) surface is a type of nanotopography that has attracted a lot of attention. NTs can be easily fabricated by anodization in an economical way, and the nanotube dimensions can be precisely controlled [19,20,21]. These Ti-based nanostructures significantly change the surface properties of Ti, regarding the morphology, topography, and wettability, as well as imprinting a slight antimicrobial character [6,12,16]. Moreover, NT topography presents similar characteristics to natural bone (pore size/diameter 60–100 nm), which may be beneficial for the bone cell interference response [22].

The dental implant is considered “open” due to the communication with the oral cavity, being exposed to the salivary environment, which is a septic environment with many aggression factors [23]. Thus, it is susceptible to contamination with biofilm-forming oral bacteria and, subsequently, with periodontal pathogens [24]. Therefore, the development of antimicrobial dental treatments is essential to prevent dental infections and early treatment failure [25]. To prevent infections, various antibacterial agents (antibiotics, oxides or metal ions, antimicrobial peptides) can be loaded on NT surfaces for local action [21,26,27].

Zinc oxide (ZnO) nanoparticles have intrinsic characteristics, such as a wide band gap, high electronic conductivity, nontoxicity, chemical durability, and high electrical, optical, piezoelectric, thermal, and mechanical properties [16,28]. Due to these properties, they are used as a drug delivery system and in the treatment of cancer, infections, diabetes, and other diseases [28]. The antimicrobial activity of ZnO nanoparticles has been demonstrated against oral bacteria (Gram-positive and Gram-negative), resulting in the bacteria death by oxidative stress (generate reactive oxygen species; ROS), protein dysfunction, and membrane and DNA damage [21,24,29]. Because of its involvement in boosting osteoblast functions and inhibiting osteoclasts, ZnO is a good candidate for Ti implant modification [14]. According to Xiaheng Wang and coworkers, a bilayer biomimetic nano-ZnO could lead to an implant material with not only high antibacterial activity against both *E. coli* and *Staphylococcus aureus* but also low cellular cytotoxicity [30]. Vancomycin (Van) was loaded into ZnO-FA (folic acid) NTs to generate a pH-sensitive NTs implant, according to Xiang et al. [14]. Furthermore, ZnO is one of the inorganic nanomaterials that the Food and Drug Administration (FDA) has approved for use on the human body [30]. It is very important to find the most suitable method for ZnO encapsulation into a coating.

Metal implant surfaces modified with polymeric coatings are considered advantageous in reducing bacterial adhesion [27]. Natural biopolymers such as collagen, chitosan, dopamine, sodium alginate, silk fibroin, and hyaluronic acid are capable of simulating the natural bone extracellular matrix [11]. These polymers can be used as coatings able to deliver active substances at the desired place [31].

Silk fibroin (SF), a natural protein spun by silkworms (Bombyx mori), is a biomaterial that has been extensively investigated for biomedical applications. It presents remarkable properties such as non-toxicity, high biocompatibility, adequate biodegradability, superior mechanical and elastic properties, and deficient inflammatory reactions [6,32,33,34,35,36]. The presence of SF improves the repair process of mineralized osteodentin tissues [34] and bone regeneration [32]. To modulate the release rates for implants, silk fibroin (SF) nanofibers electrospun onto vancomycin-loaded TNTs were used by Fathi et coworkers [14]. SF represents a favorable matrix, as it can be obtained in a variety of physical forms suitable for encapsulating active substances (antibiotics, vaccines, metal oxides, insulin, and other macromolecules) [35,37]. However, SF has no inherent antimicrobial activity, being inclined to microbial attacks, lessening its applicability [38].

Polydopamine (PD), which is an adhesive protein derived from mussels, is produced through an autopolymerization process of dopamine. PD was studied intensively due to its remarkable properties, especially for the strong adhesive properties [39,40,41,42]. PD has a strong affinity for a large variety of substrates [41,42]. Therefore, it can be used as a coating material to improve the adhesion and biocompatibility of biomaterial surfaces [36]. PDA coatings are used to bind functional substances (drugs, metal or oxide nanoparticles, hydroxyapatite, etc.) on the surface of metallic biomaterials to improve biocompatibility, osseointegration, cytotoxicity resistance, and antibacterial activity [39,42]. Drug-loaded NTs have been covered with biopolymers including polydopamine (PDA), poly(lactic-co-glycolic acid), and chitosan to achieve sustained long-term release patterns. To enhance implant osteogenesis, as-fabricated titanium nanotubes were coated with polydopamine, followed by the addition of bone morphogenetic protein-2BMP-2 by Min Lay and coworkers. The use of polydopamine-functionalized SrTiO_3_ nanotubes for combination osteoinductive and antibacterial properties was described by Qiao et al. in 2019 [14].

Combining silk fibroin with dopamine, two natural polymers, can result in a material that is biocompatible, environmentally friendly, and with enhanced mechanical properties, that can be processed into various forms (films, fibers, gels, sponges) making it suitable for a range of applications, especially in the biomedical field [43]. Moreover, dopamine can be used as a platform for further chemical modifications. The silk fibroin–dopamine combination can be easily functionalized with other active compounds for additional functionalities, such as antimicrobial properties or enhanced cell attachment [44,45].

Considering the notions presented above, the present work proposes a novel approach in creating Ti coatings with an improved antimicrobial effect, using nature-inspired biomimetic polymers. First, the surface of Ti was nanostructured with TiO_2_ nanotubes, followed by the deposition of silk fibroin fibers using an electrospun process. An important aspect of originality for this work consists of approaching and optimizing different methods for the coating functionalization process with ZnO as an antibacterial agent. This was achieved using three methods, and the resulting hybrid coating properties were evaluated comparatively to assess the morphology, topography, wettability, and antibacterial effect—properties that determine the biological behavior of the material. Furthermore, electrochemical stability, an issue not very often addressed in the literature for these types of coatings, was discussed in correlation with the chemical structure.

## 2. Materials and Methods

### 2.1. Materials

This investigation made use of titanium (Ti) pieces (1 cm × 1 cm), 99.7% purity and 0.1 cm thick, provided from Alpha Aesar (Tianjin, China). The Ti samples were cleaned with distilled water, ethanol, and acetone for 15 min before being polished using abrasive papers of various porosities.

Other reagents: methanol, ethanol, acetone, hydrochloric acid, Trizma^®^ base, dopamine hydrochloride, lithium bromide, calcium carbonate, anhydrous ethylene glycol (EG) (99.8% purity), ammonium fluoride, and ZnO nanoparticles were provided by Sigma Aldrich (Steinheim, Germany). The Bombyx mori cocoons were purchased from a local farmer. All the substances used for the antibacterial test were purchased from VWR (Bucharest, Romania).

### 2.2. Preparation of TiO_2_ Nanotube on Titanium Surface

The anodizing procedure was conducted in an electrochemical cell with two electrodes, a Ti electrode as the anode and a Pt electrode as the cathode. The voltage was adjusted from 0 to 40 V with a MATRIX MPS-7163 source with a rate of 2 V/10 s and then held constant at room temperature. The titanium samples were anodized at 40 V for 3 h in an organic electrolyte containing EG, 0.5% wt. NH_4_F, and 2% distilled water. The anodized samples are named NT [46].

### 2.3. Construction of Hybrid Film TiO_2_ Nanotubes/Silk Fibroin/Polydopamine/ZnO Nanoparticles

#### 2.3.1. Deposition of Silk Fibroin Fibers

The source of the silk fibroin was Bombyx mori cocoons. The silk fibroin extraction procedure followed the protocol described in our earlier study [47]. The concentration of the obtained silk fibroin solution after the purification process through dialysis was 6.45%, and then it was stored at 4 °C.

The precursor solution for producing silk fibroin fibers using electrospun fibers starts with the production of a silk fibroin/PEO blend by combining a 4:1 volume ratio of 6.45% (*w*/*v*) silk fibroin aqueous solution with 5.0% (*w*/*v*) polyethylene oxide (PEO). It resulted in a 3% silk fibroin/PEO solution. The PEO solution was added to the electrospun solution to increase viscosity and surface tension [48].

Then, 1 mL of this solution was introduced in a plastic syringe. A syringe pump (Legato 180 from KDS Scientific, Holliston, MA, USA) was used to apply a 0.75 mL/h flow rate of liquid. The PS/EJ30P20 high voltage DC power supply (made by Glasmann High Voltage Inc., High Bridge, NJ, USA) was used to apply 15 kV between the syringe needle tip and the grounded collector plate for 10 min. The distance between the syringe needle’s tip and the collector plate was set at 15 cm. NT samples were fixed on the collector to obtain NT/SF samples. Afterwards, the samples were submerged in 90% methanol for 15 min, followed by 24 h in ultrapure water (for PEO removal). A methanol treatment was used to induce β-sheet structure to create insolubility of the silk fibers in the aqueous solution [48].

#### 2.3.2. Deposition Silk Fibroin Fibers/ZnO Nanoparticles

First, a solution containing ZnO nanoparticles (0.02 wt.%), Triton X (0.01% vol.) (Sigma Aldrich, Saint Louis, MO, USA), and distilled water was prepared. This solution was added for a final concentration of 0.125% vol. for ZnO into the silk fibroin/PEO solution. The fibers were deposited from this mixture by the electrospun technique under the conditions described above. The resulting samples were named NT/SF-ZnO.

#### 2.3.3. Deposition of Zn Nanoparticles and Silk Fibroin Fibers

First, 100 µL was extracted from the ZnO solution that was already made. The spin coating method is used to deposit this solution on the NT surface. The NT substrate is fixed on a spin coater, and the disk speed is set to 3000 rpm for 30 s. Following that, the samples were calcined for 10 min at 180 °C. The silk fibroin fibers were deposited on these samples under the same conditions by the electrospun method. The samples were named NT/ZnO/SF.

#### 2.3.4. Deposition Silk Fibroin Fibers, Polydopamine Films, and ZnO Nanoparticles

A TRIS buffer solution at pH 8.5 was prepared from distilled water, 0.2 M Trizma^®^, and 0.2 M HCl [49]. In this solution, we dissolved the dopamine (2 g/L) [50]. The NT/SF samples were immersed in 10 mL polydopamine solution for 8 h in the dark at room temperature under constant shaking. The samples were washed using distilled water. Then, the ZnO nanoparticles were deposited on these samples using the spin coating method. They were named NT/SF/PD/ZnO.

### 2.4. Samples Characterization

Scanning electron microscopy (SEM) gave details on the morphology of the NT modified surface. Using an FEI QUANTA 650 FEG (Hillsboro, OR, USA, SEM with Field Emission Gun), ultra-high-resolution images were captured in high vacuum at various magnifications. Energy dispersive X-ray analysis (EDX, Oxford X-max 80 SDD, Thermo-Fischer Scientific, Hillsboro, OR, USA) was utilized on the ZnO samples. The sample roughness was examined using atomic force microscopy (AFM) using the A100-AFM SGS (APE Research, Trieste, Italy) equipment and Gwyddion 2.9 software (APE Research, 2.9, Trieste, Italy) for data processing. A Fourier transform infrared spectrometer (FTIR), model Spectrum 100 (PerkinElmer, Waltham, MA, USA), was used to measure the infrared spectra. The results were an average of four scans with a resolution of 4 cm^−1^ and a range of 1000 to 4000 cm^−1^. Using a CAM 100 Optical Contact Angle Meter instrument (KSV Instruments, Espoo, FIN) and the sessile drop technique, the surface’s wettability was evaluated. A Hamilton syringe was used to deposit 3–5 µL water drops on the sample surface. The measurements were carried out at room temperature using the sessile drop method. To calculate the average and standard deviation, the contact angle was measured a minimum of three times in various zones on each sample surface. An Autolab 302 N potentiostat-galvanostat (Metrohm Autolab B.V., Utrecht, The Nederlands) was used to investigate the electrochemical performance using a different method at room temperature. An electrochemical cell with three electrodes was used: a working electrode (NT modified samples), a platinum rod as a counter electrode (Metrohm, Utrecht, The Netherlands), and an Ag/AgCl 3 M KCl reference electrode (Metrohm, Utrecht, The Nederlands). The electrochemical impedance spectra (EIS) were measured in the frequency range of 0.1–10^5^ Hz and registered with an amplitude (AC) of 10 mV. The Tafel plots were recorded between −260 mV and +150 mV against open-circuit potential (OCP) with a 1 mV step and 2 mV/s scan rate. The cyclic voltammetry curves were recorded by measuring the potential between −1.5 V and 1 V with a 100 mV/s scan rate. The measurements were performed in a 0.9% NaCl solution. Nova 1.11 software was used to process all the data.

### 2.5. Antibacterial Assay

The following human pathogenic microbial strains were investigated to evaluate the antibacterial activity of the materials that were tested: *Staphylococcus aureus* ATTC 25923 and *Enterococcus faecalis* ATCC 29212 (facultative anaerobe). The bacterial strains were cultivated at 37 °C on Luria Bertani Agar (LBA) plates.

The formula established by Jaiswal et al. [51] was used to determine the antibacterial effect of the samples by computing the percentage of growth.
I % = [(B_18_ − B_0_) − (C_18_ − C_0_)]/(B_18_ − B_0_)∙100 (1)

In this context, the variable “I” denotes the percentage inhibition of growth. The term “B_18_” refers to the blank-compensated optical density at 600 nm. Similarly, “B_0_” represents the blank-compensated OD_600_ of the positive control of the organism at the initial time point (0 h). “C_18_” signifies the negative control compensated OD_600_ of the organism in the presence of the test sample at the final time point (18 h). Finally, “C_0_” denotes the negative control compensated OD_600_ of the organism in the presence of the test sample at the initial time point (0 h).

Briefly, sterile specimens were incubated for 18 h in test tubes with a 5 mL culture of Gram-positive bacteria, *Staphylococcus aureus* and *Enterococcus faecalis*. After 18 h, the optical density was measured. At 200 rpm, the Laboshake Gerhardt incubator was used to conduct the incubation. Using a Jenway UV-VIS spectrophotometer (Interworld Highway LLC, Long Branch, NJ, USA), the optical density of the samples and the control (bacteria culture without sample) was measured at 600 nm. This showed that the bacteria had grown.

## 3. Results and Discussion

### 3.1. Surfaces Morphology and Composition

The SEM results of the nanostructured titanium demonstrated that the surface is covered with nanotube structures with uniform morphology and neat arrangement. Top view and cross-section SEM images of the sample obtained by anodizing in aqueous NH_4_F/H_2_O/EG electrolyte are presented in Figure 1. In these experimental conditions, self-organized and vertically aligned nanotubes grew on the surface of Ti. In addition, the nanotubes are well-defined and uniformly distributed across the surface with an open-top morphology without the presence of nanograss, as seen in Figure 1a. They have external diameters of approximately 90 nm and thin walls (Figure 1b). The cross-section images (Figure 1c) showed nanotube lengths of approximately 7.5 µm. It is visible that nanotubes have a bamboo-like structure on the exterior.

Figure 2 shows the morphology of the electrospun SF fibers before and after the immobilization of ZnO NPs on the TiO_2_ nanotubes surface, as well as the EDX spectra for samples containing ZnO. Figure 2a shows a dense layer of interconnected smooth silk fibroin fibers in a wide range of thicknesses and lengths. High magnification revealed a diameter of 326 nm for SF fibers. The fibers were uniformly spread on top of nanotubes, resulting in a matrix that combines the microstructure of SF fibers and the nanostructure corresponding to TiO_2_ nanotubes. This is an important aspect because the modified surface did not have the same morphology, which is good for the cell’s attachment. Figure 2b,c indicate that the ZnO nanoparticles addition has no impact on the SF fibers morphology. The size of SF fibers changed after polydopamine immobilization, becoming coarser (d = 450 nm; Figure 2d). Small aggregates were deposited on the fiber surface following ZnO coating, as illustrated in Figure 2d.

The results of the EDX study of the NT/SF-ZnO, NT/ZnO/SF, and NT/SF/PD/ZnO samples are shown in Figure 2e–g. Zinc, oxygen, and titanium peaks were identified.

Table 1 presents the corresponding semi-quantitative element weight percent results from the EDX spectra for modified NT substrates containing ZnO nanoparticles. Generally, the oxide composition corresponds to the alloying element ratio. However, small levels of some elements can occasionally be found in oxide mixes. The weight % of Zn, O, and Ti elements were identified. The Zn percentage in all the three samples is between 0.37 and 0.47. This indicates that ZnO NPs were successfully embedded into the hybrid matrix.

The SEM analysis reveals that all the samples present a uniform distribution of TiO_2_ nanotubes and SF nanofibers. Therefore, the EDX analysis demonstrates the presence of Zn nanoparticles on the modified NT surface.

### 3.2. Surface Topography and Wettability

The biological processes that include bacterial or cell interaction with surfaces are influenced by the biomaterial surface wettability [52,53]. Two of the surface qualities that affect wettability are surface chemistry and topography. Understanding and managing the mechanisms of wetting are crucial. Today, the design of the implant surface is mostly based on certain micro- and nano-topographical properties. Most of the time, hydrophilic surfaces resist the attached microbe and maintain the clean surfaces.

In this study, the Sessile drop method was utilized to investigate the influence of the chemical composition of the coatings and roughness on the wettability of the nanotubular surfaces (Figure 3). The contact angles of NT and NT-SF surfaces were measured as 28°± 0.57° and 46° ± 0.99°, respectively, as was also observed in our previous research [47]. It is obvious that all the surfaces have hydrophilic character because the contact angle is <90°. The hydrophilicity of the nanotubular surface decreases when SF fibers are deposited (Figure 3). This decrease is explained based on the presence of hydrophobic domains in SF as well as its β-sheet crystal structure [52]. With the addition of ZnO, the contact angle reduced to 42° ± 0.51° in NT/SF-ZnO and 35° ± 0.9° in NT/ZnO/SF. The contact angle corresponding to NT/SF/PD/ZnO has a value of 52° ± 0.28° and is very comparable to the PD substrate that was previously published [47].

Bone–implant interface development and bacterial contact are both highly dependent on surface roughness [54,55].

Figure 4 presents 3D AFM images corresponding to all the tested samples. The roughness of the NT/SF sample decreased to 95 nm from 125 nm, which corresponds to the NT substrate, while the contact angle increased. The roughness and hydrophilicity of the samples containing SF and ZnO nanoparticles (NT/ZnO/SF and NT/SF-ZnO) are higher than those of NT/SF. Corelating this with wettability, the Wenzel model can be applied: by increasing the surface roughness, the hydrophilic substrate becomes more hydrophilic [56,57,58].

Compared with the NT/SF substrate, the NT/SF/PD/ZnO sample has a higher roughness and contact angle. This “mushroom state”, kind of wetting behavior, is an intermediate state between the Cassie–Baxter and Wenzel models [49].

The variety of morphologies might be used to explain these differences in water contact angle.

Hydrophilic surfaces generally exhibit resistance to microbial adhesion, hence, preserving their cleanliness. The investigation has demonstrated that biomaterials exhibiting higher levels of hydrophilicity facilitate enhanced cellular connections [59].

### 3.3. Fourier Transform Infrared Spectroscopy

FTIR analysis is frequently employed to investigate SF structure because amide band positions are highly susceptible to conformational changes [60]. This method was also utilized to assess the SF structure and to investigate potential interactions between the SF and PD coatings. The FTIR spectra was registered in the 1000 to 4000 cm^−1^ range.

According to the literature [49,60,61], silk protein is composed of amide groups. The specific vibration bands around 1620 cm^−1^ were attributed to the absorption peak of the peptide backbone of amide I (C = O stretching), bands around 1513 cm^−1^ to amide II (N-H bending), and bands around 1230 and 1444 cm^−1^ to amide III (C-N stretching). An intense and strong amide band is apparent at 3282 cm^−1^, showing N-H bending. These distinct absorbance peaks indicate the presence of a hydrogen-bonded N-H group. Silk fibroin includes an α-helix absorption peak at 1655 cm^−1^, random coil conformation absorption peaks at 1650 or 1645, 1550, and 1230 cm^−1^, and β-sheet absorption peaks around 1630, 1530, and 1240 cm^−1^.

In our case, the amide I, II, and III peaks at 1614, 1520, 1445, and 1230 cm^−1^ were attributed to the obtained 6.45% silk fibroin solution (Figure 5). The peaks for the NT/SF, NT/SF-ZnO, and NT/ZnO/SF samples had approximately the same values, indicating that the nanostructured surface was effectively coated with fibroin. The characteristic peaks at 1623 cm^−1^, 1520 cm^−1^, and 1225 cm^−1^ indicate that silk fibroin is in a β-sheet conformation. The presence of ZnO nanoparticles does not influence the FTIR spectrum.

According to our results [49], polydopamine has peaks at 3350 cm^−1^ (OH and NH stretches), 1283 and 1631 cm^−1^ (stretching vibration of catechol hydroxyl C-O and C=O), and 1523 cm^−1^ (C = C bonds in the indole structure). Because PD and SF are natural polymers, their peaks are comparable (1631 and 1523 cm^−1^). Consequently, there are no significant changes in the FTIR peak between the NT/SF and NT/SF/PD/ZnO samples, indicating that both polymers were deposited on the nanotubular surface.

### 3.4. Electrochemical Characterization

#### 3.4.1. Electrochemical Impedance Spectroscopy (EIS)

EIS was used to examine the samples’ electrochemical (corrosion parameters, charge transfer, electric double layer structure) and frequency-dependent transport properties at electrolyte–biomaterial interface processes [49,62]. The Nyquist diagrams for untreated NT and coated samples are shown in Figure 6a. For EIS data fitting, two types of electric circuits were proposed, as seen in Figure 6b,c.

The equivalent circuit shown in Figure 6b was used for NT, NT/SF, NT/SF-ZnO, and NT/ZnO/SF samples. This circuit includes the resistance of the physiological serum, NaCl 0.9% (R_solution_), in series with a compact oxide circuit (constant phase element—CPE_oxide_ and Roxide). These circuits connect in series with another that corresponds to hybrid coatings (CPE_coating_ and R_coating_). For the NT/PD/SF/ZnO sample, the circuit was updated with new elements: a resistance for polydopamine film (R_PD_) in parallel with a CPE_PD_, as seen in Figure 6c. Surface roughness, a non-homogeneous reaction rate on a surface, a coating with a variable thickness and/or composition, or an uneven distribution of current on the electrode surface are a few examples of situations where CPE might occur.

It can be observed from the data presented in Table 2 that the values for Rsol are similar for all tasted samples and have values between 60 and 100, the same electrolyte solution being used for all tests.

Compared to the pristine NT surfaces, the modified NT surfaces possess a higher resistance of the compact oxide layer (R_oxide_), as shown in Table 2. These values demonstrate an excellent relationship with the Tafel diagram results, which show that the corrosion rates of the modified NT samples were lower than those of the pristine NT sample.

When ZnO nanoparticles are added (NT/ZnO/SF and NT/SF-ZnO), the barrier oxide layer’s resistance increases compared to the NT/SF sample.

In other study, the resistance value of pure TiO_2_ nanostructure arrays decreased after direct modification with ZnO by the hydrothermal method [63]. In our study, the second resistance associated with the coating (R_coating_) is higher for all modified NT samples compared to the pristine NT substrate. A possible explanation could be that SF and ZnO nanoparticles are deposited at the surface of the nanotube layer. Moreover, it can be observed that the NT/SF-ZnO and NT/ZnO/SF samples presented higher coating resistances than the NT/SF substrate, being more stable.

NT/SF/PD/ZnO has a smaller semicircle diameter in the Nyquist plot compared to other modified NT samples. This suggests a reduced charge transfer resistance and a significantly increased charge transport at the electrode/electrolyte contact [64]. A decrease in resistance for a NT sample having a biomolecule attached through PD was also observed in a previous study [53].

Y_0_, the admittance (1/|Z|) at 1 rad/s, and N, an empirical constant with a range of 0 to 1 (when N = 1, the CPE behaves as a pure capacitor, whereas when N = 0, the CPE behaves as a pure resistor), are the two parameters that define CPE [53]. Most samples have N values for the oxide layer and surface coating in the range of 0.61 to 0.98, indicating pseudocapacitive behavior.

The EIS analysis revealed that all modified NT samples are more stable than the pristine NT and exhibit pseudo-capacitive behavior.

#### 3.4.2. Tafel Analysis

Electrochemical studies were performed on all the modified samples to determine the corrosion behavior. The Tafel diagrams (Figure 7) were used to obtain the electrochemical parameters for the investigated surfaces: corrosion potential (E_corr_), corrosion current (i_corr_), and corrosion rate (v_corr_).

The corrosion parameters were obtained by extrapolating anodic and cathodic curves (Table 3). Furthermore, the protection efficacy was calculated using the formula [65]:(2)$$PE=\frac{i_{corr(unmodified)}-i_{corr(modified)}}{i_{corr(unmodified)}}\cdot 100$$
where *i_corr_* is the corrosion density current.

Figure 7 demonstrates how similar all the polarization curves were to one another. This implied that all the NT/SF-ZnO, NT/ZnO/SF, and NT/SF/PD/ZnO composite coatings in this investigation used a similar polarization mechanism.

In general, a material with a lower corrosion current density and a higher corrosion potential has a reduced propensity to corrode and, hence, performs better against corrosion [66].

The corrosion potential was moved to positive values for NT/SF and all modified ZnO samples compared to pristine NT, as is shown in Table 3. It is evident that the surface modification with silk fibroin caused a decrease in the corrosion potential from −0.161 V for NT to more electropositive values, up to −0.071 V for NT/SF/PD/ZnO, demonstrating an increase in NT surface stability following the coating deposition.

Similarly, the density of corrosion currents was found to be lower for the coated samples compared to NT, suggesting the same thing. Therefore, the corrosion current density decreased from 3.89 × 10^−6^ A/cm^2^ for NT to values 6–9 times lower for modified samples.

Compared to pristine NT, all the modified samples containing silk fibroin and ZnO nanoparticles exhibit better electropositive potentials. The samples NT/ZnO/SF and NT/SF-ZnO exhibit a decreased susceptibility to corrosion compared to the NT samples due to lower corrosion currents. The NT/ZnO/SF sample showed a lower corrosion rate (0.0014 mm/year) compared to NT sample (0.452 mm/year).

NT/ZnO/SF presents the highest corrosion protection efficiency (97%). Probably, the ZnO nanoparticles deposited on nanotube walls can passivate the TiO_2_ surface defect states.

All coatings contributed to the increased corrosion stability and highest corrosion protection efficiency compared to pristine NT.

#### 3.4.3. Cyclic Voltammetry

The double-layer capacitance (C_dl_) may be a fundamental structural characteristic for surface interactions, being determined by the relationship between the surface charge and the oxide–solution interface [49,67]. Previous research has demonstrated that electrical double layer interactions play an important role in the adherence of bacteria. Under physiological conditions, the phosphate and carboxyl groups on the surface of bacterial cells give them a negative charge. At a physiological pH, titanium surfaces generate an electrical repulsion in the double layer when bacteria approach them due to the negative charge of the oxide on the surface [68,69,70].

The titanium nanotube electrodes’ regular pore size and hollow tube morphology make it simple to transfer charges and boost interfacial ion movement, which leads to improved capacitive performance [71]. In the nanotube layer, the open pore mouth structure helps make electrolyte ion diffusion possible [72]. The growth of the silk fibroin layer on the NT surface led to the smoothing of the surface, as observed in the AFM analysis, and the covering of some nanotubes, as seen from SEM images. Thus, the active surface area decreases, and the double layer capacity is reduced for NT/SF compared with NT (Figure 8).

By ZnO depositing on the NT/SF substrate, Figure 8, the sample surface determines an increase in the active surface and Faradaic current, as well as C_dl_ (from 1.39 to 2.06 or 1.44 mF/cm^2^). Thus, NT/SF surface modification with ZnO nanoparticles improved the surface’s hydrophilicity, wettability, and roughness, thus, speeding up the rate of charge transfer at the interface and the electrical double layer formation. The porous structure of both NT/ZnO/SF and NT/SF-ZnO can be especially attributed to charge transfer and ion diffusion, which have a significant impact on its electrochemical capacitance. The NT/SF/PD/ZnO sample has the highest C_dl_ value (10.03 mF/cm^2^), efficient ion absorption/desorption process at the interface between this surface and the electrolyte solution. The C_dl_ value obtained for the NT/SF/PD/ZnO sample is comparable to the values obtained for TiO_2_ nanotubes boron-doped diamond nanostructures from other studies [73].

The increased C_dl_ values for samples containing ZnO on top of nanotubes are in line with observations made in other studies regarding pseudocapacitive oxides coated on the top of nanotubes. Despite ZnO’s weak electrical conductivity and low specific capacitance [74], higher stability results from greater pseudocapacitive oxide adhesion to TiO_2_ substrates, and the open porous electrode morphology offers readily accessible areas for electrochemical processes and smooth routes for charge carrier transport, increasing specific capacitance [71].

#### 3.4.4. Mott–Schottky

The Mott–Schottky analysis is heavily employed for semiconductor surfaces, the capacitance changes being related to the electronic properties at the electrolyte/oxide interface.

Figure 9 shows Mott–Schottky plots of the modified or unmodified NT substrate exhibiting a single positive slope, which indicates the n-type conductivity. The voltage that is used to smooth out band bending at the semiconductor/electrolyte interface is known as the flat-band potential (Fb). The flat band potential for n-type semiconductors can be thought of as the conduction band potential [62]. The flat band potential was calculated from the X-axis intercept in the linear region [75]. The flat band potential change after surface modification could be caused by trapped electrons or holes at the interface states and fixed charge. The flat band potential increases from 0.016 V for the pristine NT substrate to 0.233 V for NT/SF. Furthermore, the charge carrier density (ND) values are increased form 2.5 × 10^17^ for NT to 7.5 × 10^17^ for NT/SF. The ND value obtained in this study is close to the one obtained in our previous study [53].

By depositing ZnO nanoparticles on the NT/SF surface, there is a decrease in the Fb value, from 0.233 V for NT/SF to 0.06 V for NT/SF-ZnO and 0.09 V for NT/ZnO/SF, suggesting a better charge separation. A similar behavior was observed in other studies when TiO_2_-NTs was coated with nanostructured ZnO [76,77]. ND is higher for hybrid nanostructures NT/SF-ZnO (3.1 × 10^17^) and NT/ZnO/SF (2.8 × 10^17^) compared to pristine NT. The rise in ND may be due to more defects being created by the addition of ZnO to the TiO_2_ matrix, which enhances the charge transfer of the nanostructures [78].

The NT/SF/PD/ZnO sample had a negative value for ND. The drop in charge transfer resistance is caused by the flat-band potential’s shift to the negative, which suggests a reduced energy barrier for interfacial electron transfer [62]. These findings are in line with those of the EIS analysis. Lower R_coating_ and R_oxide_ for NT/SF/PD/ZnO compared to those corresponding to NT/SF-ZnO and NT/ZnO/SF were a result of the improvement in charge separation. NT/SF/PD/ZnO sample was also presenting an increased corrosion rate compared to NT/SF-ZnO and NT/ZnO/SF from Tafel plots.

### 3.5. In Vitro Antibacterial Assay

Among the 700 species of bacteria that have been detected in the human mouth, the most common are *Streptococcus*, *Staphylococcus*, *Enterococcus*, *Neisseria*, *Veillonella*, *Actinomyces,* and other genera [79]. Many of these bacteria are harmless and even beneficial, but some can be pathogenic and cause conditions such as dental caries, gingivitis, periodontitis, and tooth loss [80].

For testing the antibacterial effect of the newly created surfaces, two of these bacteria were chosen: *Staphylococcus aureus* and *Enterococcus faecalis*.

*S. aureus* is a type of bacteria that is often found on the skin and in the nasal passages of healthy individuals. *S. aureus* can cause various types of oral infections, including abscesses, periodontitis, and other forms of gum disease. This can lead to symptoms such as swollen, red, and tender gums, bleeding gums, loose teeth, and bad breath. Furthermore, *S. aureus* is known for its ability to form biofilms, which are structured communities of bacteria that can adhere to surfaces and resist antimicrobial treatments. Many strains of *S. aureus* are resistant to commonly used antibiotics (such as methicillin-resistant *S. aureus*, or MRSA), making infections caused by these strains difficult to treat [81].

*E. faecalis* is a bacterium that is commonly found in the human gut, but it can also inhabit various other sites in the body, including the oral cavity. Like *S. aureus*, *E. faecalis* can form biofilms. In the oral cavity, these biofilms can be particularly difficult to remove once established. *E. faecalis* is known to be highly resistant to many commonly used antibiotics, which complicates treatment when it causes an infection. It can acquire and transfer resistance genes, contributing to the growing issue of antibiotic resistance [82].

Considering that both bacteria are already resistant to common antibiotics, the choice for using ZnO nanoparticles as an antimicrobial agent seems to be the most appropriate. Furthermore, we will discuss comparatively the results from antibacterial tests, in function of the applied method to incorporate ZnO. Furthermore, it is important to notice that the inhibition of growth increased in two steps.

It is already known from the literature that nano-structuring the titanium surface with nanotubes leads to a slight increase in the inhibition of growth compared with pristine titanium because of the great hydrophilicity of the nanotubes, and the modest number of negative charges on the surface may help to hinder the attachment of bacteria through the property of non-specific protein adsorption [83]. Thus, the importance of surface morphology could be already highlighted with this first level of surface modification. Herein, the NT sample is used as reference to report the results for the hybrid films.

When the TiO_2_ nanostructures are covered with SF fibers, the inhibition of growth continued to slightly increase for both tested bacteria. Although, from the literature, it is already known that SF lacks the antimicrobial property [84], in our study, a supplementary inhibition of growth was obtained for the NT/SF coated samples. Compared to the NT sample, the surface roughness did not vary, only the wettability decreased. Thus, this change could be associated with surface chemistry due to the interaction between the charged functional groups of silk fibroin and the bacterial cell surface.

The presence of encapsulated ZnO nanoparticles led to an inhibition of growth that increased considerably. For the case when ZnO was fixed on the NT nanotubes before SF deposition, the results for the antimicrobial tests are very comparable with the situation when the functionalization of the coating with ZnO is made at the same time with SF electrospun. The ZnO nanoparticles and the bacterium have opposite charges, which results in electrostatic forces that firmly adhere the nanoparticles to the bacteria’s surface and harm the cell membrane [85].

An interesting observation is that in the case of using polydopamine as an intermediate layer for embedding ZnO, the inhibition of growth was higher, up to 55% and 53% respectively, for both microorganisms as seen in Table 4. This was an expected difference considering the following aspects: (1) the percentage of Zn is superior compared to the other two coatings; (2) the presence of polydopamine changes the surface properties in a more pronounced manner: the surface chemistry, topography, and wettability. Thus, this result is probably a cumulative effect.

Zinc’s antimicrobial activity is thought to work through several mechanisms:
(a)High concentrations of zinc can disrupt the balance of metal ions in the bacterial cells, interfering with essential biochemical processes [86].(b)Zinc ions can inhibit certain bacterial enzymes, impeding bacterial growth and metabolism [87](c)Some studies suggest that zinc ions may destabilize bacterial cell membranes, leading to cell lysis and death [87,88].(d)Zinc is shown to inhibit the formation of biofilms, which are communities of microorganisms that can adhere to surfaces and are often resistant to antimicrobial agents [89].

It is important to note that the effect of zinc on bacteria can be species specific. Some bacteria, for example, have mechanisms to regulate zinc uptake and efflux, enabling them to survive in environments with varying zinc concentrations [86]. Zinc is used in various applications due to its antimicrobial properties, such as in certain types of lozenges for reducing the duration of common cold symptoms [85]. In dentistry, zinc oxide is often used in dental fillings and cements, where it can provide some degree of antimicrobial activity [90].

Moreover, high levels of zinc can be toxic to human cells, so any use of zinc as an antimicrobial must balance its potential benefits against the risk of toxicity [91].

The presence of *E. faecalis* and *S. aureus* in root canals was documented in therapy-resistant periapical periodontitis, and *E. faecalis* is often detected in root canal-treated teeth in values ranging from 30 to 90% of the cases. Both bacteria can cause periapical periodontitis. When compared to cases of original infection, the probability of E. faecalis being present in root canal-treated teeth is approximately nine times higher [92,93,94,95].

## 4. Conclusions

TiO_2_ nanotubes (NT) and silk fibroin /ZnO coated TiO_2_ nanotubes were prepared for this study, using various methods for ZnO nanoparticles embedding into the hybrid film. NT has uniform morphology and neat arrangement and is free of nanograss, with thin walls and external diameters of approximately 90 nm. According to SEM examination, the SF nanofibers were uniformly distributed. FT-IR spectra showed characteristic peaks proving that the nanostructured surface was effectively coated with fibroin. EDX analysis was conducted to highlight the presence of ZnO nanoparticles on the NT/SF-ZnO, NT/ZnO/SF, and NT/SF/PD/ZnO modified NT surface.

The contact angle measurements are crucial for coating implants because they help to precisely characterize the surfaces of the solid materials. All samples have contact angle < 90°, being hydrophilic. The hydrophilicity of the nanotubular surface decreases when SF fibers are deposited. The NT/ZnO/SF and NT/SF-ZnO samples had higher roughness and hydrophilicity than NT/SF. The highest contact angle and a higher roughness were observed for NT/SF/PD/ZnO compared to NT/SF.

As corrosion is an important aspect for an implant material, different electrochemical studies were performed on the studied samples such as EIS, Tafel, and cyclic voltammetry. According to the EIS study, all the modified NT samples behave in a pseudo-capacitive manner and are more stable than the original NT. In comparison to pristine NT, all coatings showed an increased corrosion stability and provided the best level of corrosion protection (over 80%) according to the Tafel plots. From cyclic voltammetry, C_dl_ values were determined. They are higher for samples with ZnO. By depositing ZnO nanoparticles on the NT/SF surface, there is a decrease in the flat-band potential value. Charge carrier density was higher for hybrid nanostructures NT/SF-ZnO and NT/ZnO/SF compared to pristine NT. A greater ZnO pseudocapacitive oxide adherence to TiO_2_ nanotubes leads to increased stability, and open porous surface morphology provides easily accessible areas for electrochemical reactions and smooth channels for charge carrier movement, boosting specific capacitance.

In vitro, the antibacterial effect was tested against *E. faecalis* and *S. aureus*. Encapsulated ZnO nanoparticles caused a growth inhibition that was far more pronounced. The antimicrobial test findings for the scenario where ZnO was fixed on the NT nanotubes prior to SF deposition are quite equivalent to those for the case when the functionalization of the coating with ZnO is made concurrently with SF electrospun. An intriguing finding is that the growth inhibition was greater for both microorganisms when polydopamine was used as the intermediate layer for embedding ZnO, reaching up to 55% against *E. faecalis* and 53% against *S. aureus*, respectively.

This study paves the path for the use of nano-engineered structures (nanotubes), together with the natural, non-toxic features of silk fibroin and dopamine polymers in anti-corrosion and antibacterial nanocomposite coatings for titanium metallic implant surface modification.

## Figures and Tables

**Figure 1 materials-16-05855-f001:**
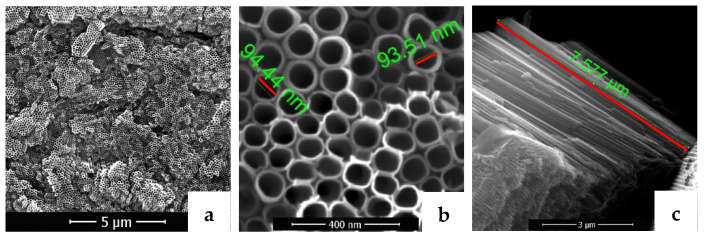
SEM images: (**a**) represents NT sample at a magnitude of 5000×, (**b**) represents the top view of NT sample at a magnitude of 100,000×, and (**c**) represents the cross-section at a magnitude of 15,000×.

**Figure 2 materials-16-05855-f002:**
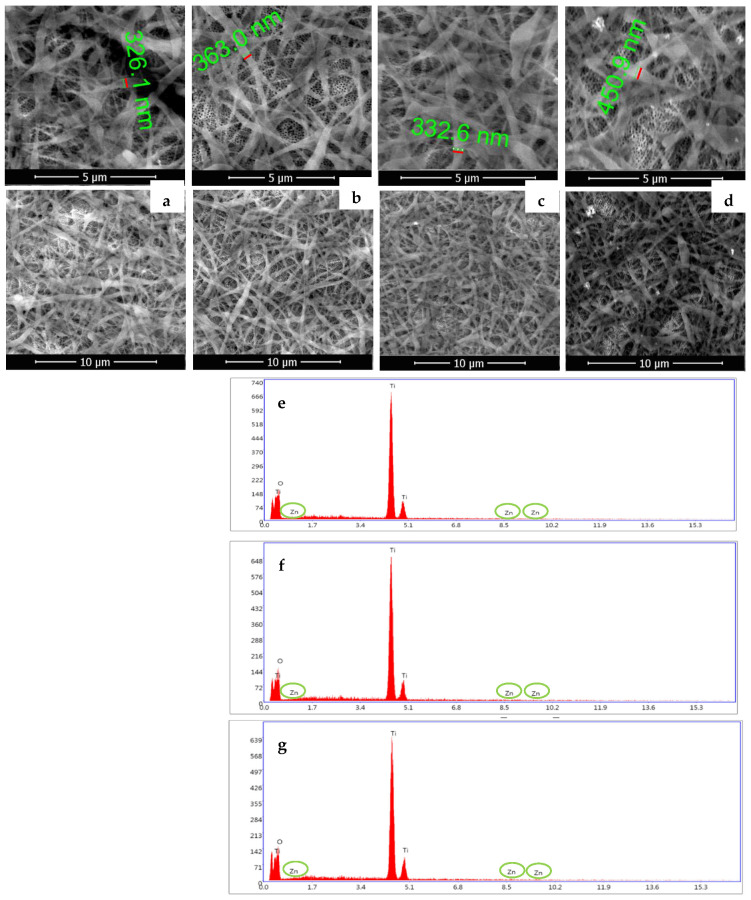
SEM images corresponding to (**a**) NT/SF (magnitude of 8236× and 4030×); (**b**) NT/SF-ZnO (magnitude of 7011× and 3829×); (**c**) NT/ZnO/SF (magnitude of 9006× and 3689×); (**d**) NT/SF/PD/ZnO (magnitude of 8806× and 4509×); and corresponding EDX diagrams for ZnO containing samples: (**e**) NT/SF-ZnO, (**f**) NT/ZnO/SF, and (**g**) NT/SF/PD/ZnO.

**Figure 3 materials-16-05855-f003:**
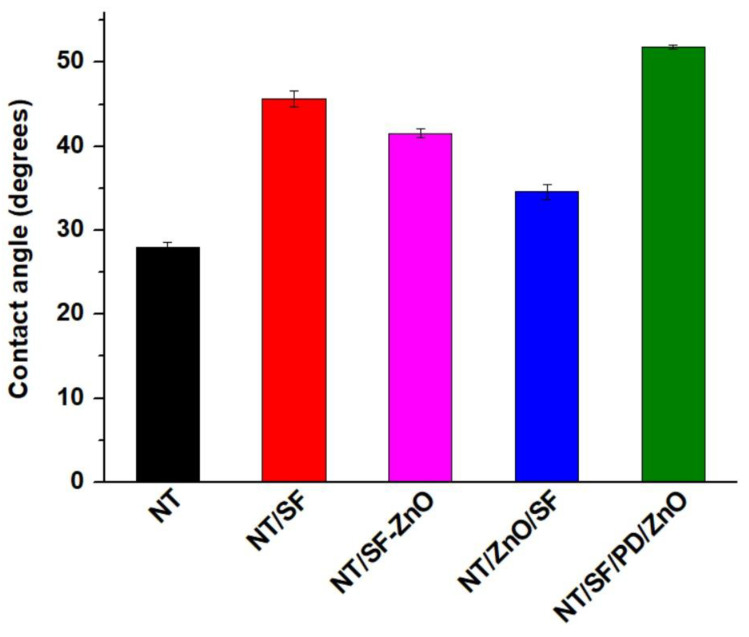
Surface wettability investigation by the Sessile drop method for all samples.

**Figure 4 materials-16-05855-f004:**
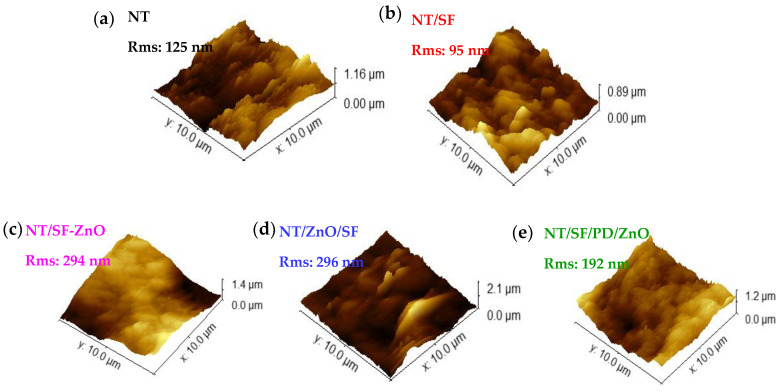
AFM 3D images, roughness values for: (**a**) NT; (**b**) NT/SF; (**c**) NT/SF-ZnO; (**d**) NT/ZnO/SF; (**e**) NT/SF/PD/ZnO.

**Figure 5 materials-16-05855-f005:**
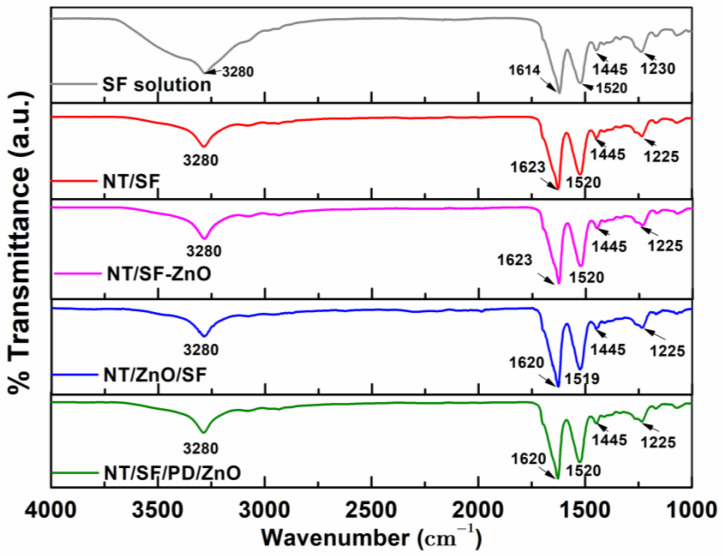
FT-IR images corresponding to the studied samples.

**Figure 6 materials-16-05855-f006:**
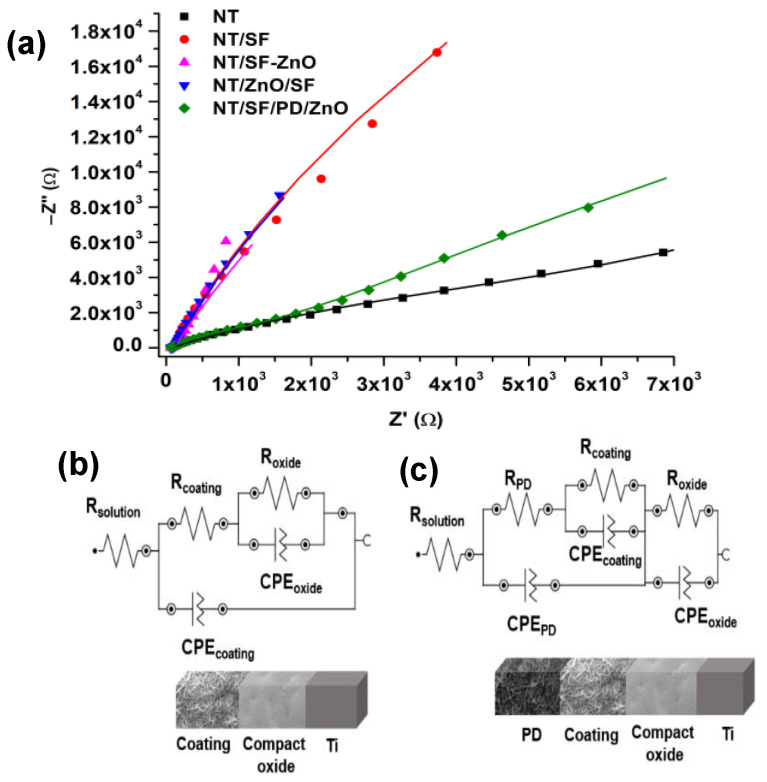
(**a**) Nyquist diagrams for the studied samples; (**b**) equivalent circuits used to fit the EIS data for NT, NT/SF, NT/SF-ZnO, and NT/ZnO/SF; (**c**) equivalent circuits used to fit the EIS data for NT/SF/PD/ZnO.

**Figure 7 materials-16-05855-f007:**
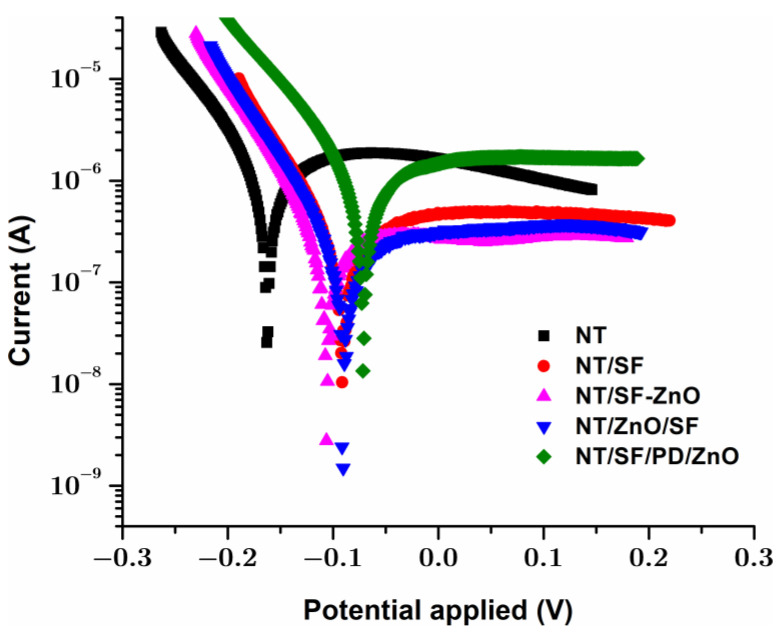
Tafel diagram for the studied samples, recorded in NaCl 0.9%.

**Figure 8 materials-16-05855-f008:**
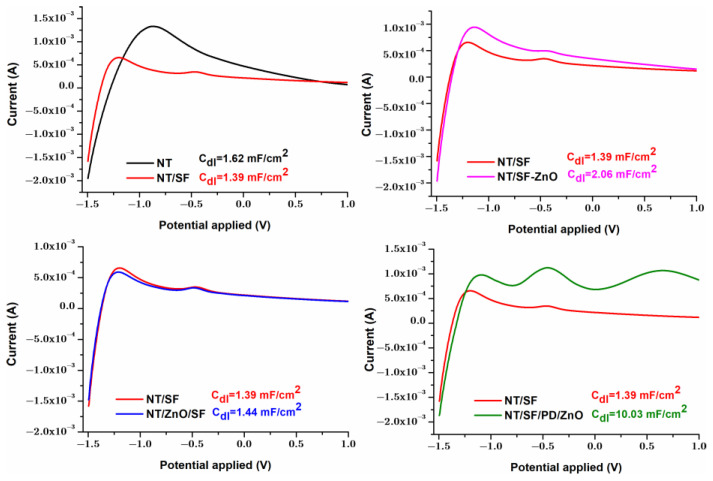
Anodic branch of the cyclic voltammograms for all the tested samples, used for Cdl calculation.

**Figure 9 materials-16-05855-f009:**
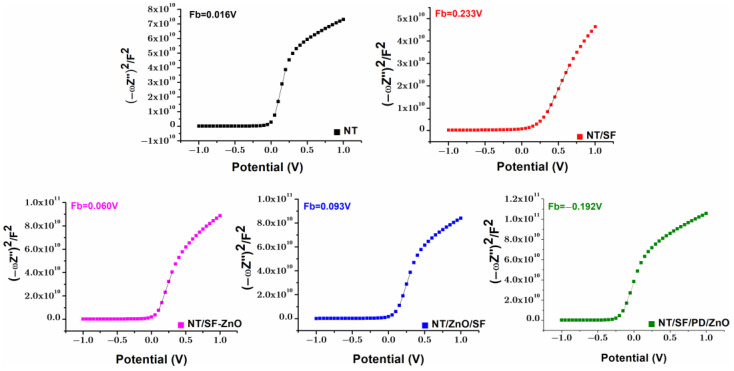
Mott–Schottky plots for all tested samples.

**Table 1 materials-16-05855-t001:** EDX analysis of TiO_2_ nanotube arrays and ZnO nanoparticles.

Samples	Weight %		
	Ti	O	Zn
NT/SF-ZnO	52.46	47.17	0.37
NT/ZnO/SF	54.32	45.28	0.40
NT/SF/PD/ZnO	52.18	47.35	0.47

**Table 2 materials-16-05855-t002:** Values obtained from fitted EIS data using NOVA software.

Parameters Samples	R_S_ (Ω)	R_PD_ (Ω)	CPE_PD_		R_coating_ (Ω)	CPE_coating_		R_oxide_ (Ω)	CPE_oxide_		X^2^
			Y_0_ (µS·s^n^)	N		Y_0_ (µS·s^n^)	N		Y_0_ (µS·s^n^)	N	
NT	99				4 × 10^3^	62	0.71	20 × 10^3^	149	0.68	0.01
NT/SF	91	-	-	-	8 × 10^3^	266	0.92	136 × 10^3^	65	0.96	0.07
NT/SF-ZnO	68				9 × 10^3^	278	0.82	156 × 10^3^	900	0.93	0.04
NT/ZnO/SF	68				10 × 10^3^	356	0.87	184 × 10^3^	897	0.98	0.03
NT/SF/PD/ZnO	63	1.7 × 10^3^	79	0.74	5 × 10^3^	270	0.61	92 × 10^3^	297	0.89	0.01

**Table 3 materials-16-05855-t003:** Corrosion parameters for studied samples.

Samples	Corrosion Potential (V)	Corrosion Current Density (A/cm^2^)	Corrosion Rate (mm/Year)	Protection Efficiency (%)
NT	−0.161	3.896 × 10^−6^	0.0452	-
NT/SF	−0.092	0.359 × 10^−6^	0.0042	91
NT/SF-ZnO	−0.104	0.249 × 10^−6^	0.0029	94
NT/ZnO/SF	−0.091	0.119 × 10^−6^	0.0014	97
NT/SF/PD/ZnO	−0.071	0.568 × 10^−6^	0.0066	84

**Table 4 materials-16-05855-t004:** Antibacterial effect for the modified Ti surfaces with NT, SF, and ZnO.

Samples	*Enterococcus faecalis*	*Staphylococcus aureus*
NT	-	-
NT/SF	15	13
NT/SF-ZnO	41	39
NT/ZnO/SF	42	41
NT/SF/PD/ZnO	55	53

## Data Availability

Data sharing is not applicable to this article.

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
