# Peer review of "Silk Fibroin/ZnO Coated TiO2 Nanotubes for Improved Antimicrobial Effect of Ti Dental Implants"

_materials, 2023, doi:10.3390/ma16175855_

Round 1

Reviewer 1 Report

Review: Manuscript ID: materials-2550666 

Title: Silk fibroin /ZnO coated TiO2 nanotubes for improved antimicrobial effect of Ti dental implants

Here are some specific considerations:

 1. The introduction need to add more detail about

- What is the main question addressed by the research?

- Why do the authors select silk fibroin combine with polydopamine?

- What does it add to the silk fibroin combine with polydopamine used in the mixture compared with other publication?

- Review of other research related to the study and get to the points of experiment.

 2. The abstract need to add more detail about results and discussion such as the evaluation and comparison with other publication?

 3. Materials and methods section are well written with citation.

 4. In the result section

·       Please clarify all Figure caption. For example, Figure 1, in my opinion, the authors may put a), b), c) in each picture and elaborate about a) is the picture represent sample………..magnitude of …..X, b) represent the top view of sample……… with magnitude of ……X and c) represent the cross section………….. for better understanding.

·       And also in Figure 2, the authors should add the magnitude scale (X) of the top and the bottom picture for better understanding.

·       Please clarify the SEM figures. Why did the authors present the SEM? I suggest that the authors needed to add circle or arrow or point out what the area the authors indicate that it is the length of nanotube? (The green line might hardly observe in the picture).

·       In Table 1, the EDX analysis of TiO2 nanotube arrays and ZnO, the authors assume that the percentage of Zn is not different. However, there is no standard deviation (SD) to show the significantly different. The sample NT/SF-ZnO shows %Zn = 0.37 and the sample NT/SF/PD/ZnO expresses %Zn = 0.47 which 0.1% different. Does this information affect to the anti-microbial? Please explain.

·       According to the authors used Ti coating, does the authors perform percentage of drug release?

 5. Conclusion section needs improvements, it looks very much general, make it little scientific with some modification.

Minor editing of English language required

Author Response

Answers to Editor and Reviewers’ suggestions

The authors thank the Reviewers and Editor for thoughtful critiques of our manuscript: "Silk fibroin /ZnO coated TiO2 nanotubes for improved antimicrobial effect of Ti dental implants" (Manuscript ID: materials-2550666) (authors: Angela Gabriela Păun, Cristina Dumitriu, Camelia Ungureanu, Simona Popescu), and for their suggestions and for helpful comments that will greatly improve the manuscript.

            As indicated below, we have checked all the comments provided by the Reviewers and we have made necessary changes according to their indications.

            All the modifications are yellow highlighted in the revised manuscript.

            Our responses to Reviewers comments are detailed on the following pages.

            The suggestions/comments of the Reviewers are in black, and our responses are in blue.

Thank you very much for your kind assistance!

Sincerely yours,

Assoc. Prof. Dr. Eng. Simona Popescu

Response to Reviewer 1 Comments

Title: Silk fibroin /ZnO coated TiO2 nanotubes for improved antimicrobial effect of Ti dental implants

Here are some specific considerations:

Question 1.

The introduction needs to add more detail about

Answer 1: Thank you very much for your suggestions! The Introduction section was completed with more details, according to your observations.

  1. What is the main question addressed by the research?

We added more information in the Introduction section:

“There have been progressive implant failures associated with peri-implant illnesses (peri-mucositis and peri-implantitis) [14]. Because systemic dosages may not be effective in overcoming the constraints of traditional therapy, local injection of medications to the bone is also acknowledged as a possible alternative [15]. Therefore, surface modification for titanium is frequently necessary to enhance its bio-performance as implant [11,12,16-18]. Choosing biomimetic dental implants with antibacterial properties (in such a way to prevent multidrug resistance) in those with a history of periodontitis may help avoid peri-implantitis, avoiding recurring visits to the doctor for debridement and significantly decreasing expenses and patient suffering [14].

Considering the notions presented above, the present work proposes a novel approach in creating Ti coatings with improved antimicrobial effect, using nature-inspired biomimetic polymers. “

  1. Why do the authors select silk fibroin combine with polydopamine?

We added more information in the Introduction section:

“Combining silk fibroin with dopamine, two natural polymers, can result in a material that is biocompatible, environmentally friendly, with enhanced mechanical properties, that can be processed into various forms (films, fibers, gels, sponges) making it suitable for a range of applications, especially in the biomedical field.  Moreover, dopamine can be used as a platform for further chemical modifications. The silk fibroin-dopamine combination can be easily functionalized with other active compounds for additional functionalities, such as antimicrobial properties or enhanced cell attachment […]”

  1. What does it add to the silk fibroin combined with polydopamine used in the mixture compared with other publications?

An important aspect of originality for this work consists in approaching and optimizing different methods for coating functionalization process with ZnO, as antibacterial agent embedded in biopolymeric matrix.

Furthermore, electrochemical stability of the new coating, an issue not very often addressed in literature for these types of coatings, was discussed in correlation with the chemical structure.

  1. Review of other research related to the study and get to the points of experiment.

According to Xiaheng Wang and coworkers, a bilayer biomimetic nano-ZnO could lead to an implant material with not only high antibacterial activity against both E. coli and Staphylococcus aureus, but also low cellular cytotoxicity [30]. Vancomycin (Van) was loaded into ZnO-FA (folic acid) TNTs to generate a pH-sensitive TNTs implant, according to Xiang et al [14].

Drug-loaded TNTs have been covered with biopolymers including polydopamine (PDA), poly(lactic-co-glycolic acid), and chitosan to achieve sustained long-term release patterns. To enhance implant osteogenesis, as-fabricated titanium nanotubes were coated with polydopamine, followed by the addition of bone morphogenetic protein-2BMP-2 by Min Lay and coworkers. The use of polydopamine-functionalized SrTiO3 nanotubes for combination osteoinductive and antibacterial properties was described by Qiao et al. in 2019 [14].

To modulate the release rates for implants, silk fibroin (SF) nanofibers electrospun onto vancomycin-loaded TNTs were used by Fathi et coworkers.

 Question 2. The abstract need to add more detail about results and discussion such as the evaluation and comparison with other publication?

Answer 2: We made improvements for the abstract section. Thank you very much for your suggestions!

 Question 3. Materials and methods section are well written with citation.

Answer 3: Thank you very much for your appreciation! References regarding the synthesis of TiO2 nanotubes and the deposition of dopamine by polymerization have been included in the text.

 Question 4. In the result section

  1. Please clarify all Figure caption. For example, Figure 1, in my opinion, the authors may put a), b), c) in each picture and elaborate about a) is the picture represent sample………..magnitude of …..X, b) represent the top view of sample……… with magnitude of ……X and c) represent the cross section………….. for better understanding.

Answer 4a: Figure 1 was detailed according to the requirements (the SEM images were marked with a, b and c, respectively their magnifications).

  1. And also in Figure 2, the authors should add the magnitude scale (X) of the top and the bottom picture for better understanding.

Answer 4b: Figure 2 has been updated by adding the magnitude scale (X) of the top and the bottom images.

  1. Please clarify the SEM figures. Why did the authors present the SEM? I suggest that the authors needed to add circle or arrow or point out what the area the authors indicate that it is the length of nanotube? (The green line might hardly observe in the picture).

Answer 4c: All SEM images have been improved according to the suggestions. The diameters and length of the NT, as well as the nanofibers dimensions, are highlighted now with a red line. The green line is coming from SEM soft.

  1. In Table 1, the EDX analysis of TiO2 nanotube arrays and ZnO, the authors assume that the percentage of Zn is not different. However, there is no standard deviation (SD) to show the significantly different. The sample NT/SF-ZnO shows %Zn = 0.37 and the sample NT/SF/PD/ZnO expresses %Zn = 0.47 which 0.1% different. Does this information affect to the anti-microbial? Please explain.

Answer 4d: According to the EDX results and the antibacterial tests, it can be observed that as the percentage of ZnO increases (0.37% for the NT/SF-ZnO sample and 0.47% for the NT/SF/PD/ZnO sample) there is also an increase in the effect antibacterial (for E. faecalis from 41% to 55% and for S. aureus from 39% to 53%). Thus, considering that the amount of ZnO nanoparticles is higher for the NT/SF/PD/ZnO sample, the antibacterial effect also increases.

  1. According to the authors used Ti coating, does the authors perform percentage of drug release?

 Answer 4e: Thank you for your comments! We agree that drug release is a very important aspect for this type of application, and it will be considered for a future study, together with in vitro biocompatibility testing.

Question 5. Conclusion section needs improvements, it looks very much general, make it little scientific with some modification.

Answer 5: We restructured the Conclusion Section. Thank you very much for your suggestions!

Question 5. Minor editing of English language required

Answer 5: Thank you very much for your appreciation! We made the language edits in the entire manuscript!

Reviewer 2 Report

This paper reports the development of Silk fibroin/ZnO coated Ti nanotubes for improvement of the antimicrobial effect of the TI implant. The NT/SF/PD/ZnO coated Ti was characterized surface analysis, electrochemical assessments, and antibacterial assay. The developed NT/SF/PD/ZnO coated Ti was outstanding properties and is potential application for dental implant. The reviewer comments are listed as follows.

1. Biocompatibility

In general, Titanium implant must be biocompatible. In vivo and in vitro experiments using appropriate cells or animals should be needed to verify the novel material's biocompatibility. At least, please discuss the biocompatibility of the material.

2. EDX spectra

For surface composition of each sample, only compositional value are listed in Table 1. If possible, please add EDX spectra for each sample. The EDX spectra can clarify the surface composition.

3. Antimicrobial assessment.

Antimicrobial assay has been performed only at the time 18 hours culture. If possible, please evaluate antimicrobial time dependence.

Author Response

Answers to Editor and Reviewers’ suggestions

The authors thank the Reviewers and Editor for thoughtful critiques of our manuscript: "Silk fibroin /ZnO coated TiO2 nanotubes for improved antimicrobial effect of Ti dental implants" (Manuscript ID: materials-2550666) (authors: Angela Gabriela Păun, Cristina Dumitriu, Camelia Ungureanu, Simona Popescu), and for their suggestions and for helpful comments that will greatly improve the manuscript.

            As indicated below, we have checked all the comments provided by the Reviewers and we have made necessary changes according to their indications.

            All the modifications are yellow highlighted in the revised manuscript.

            Our responses to Reviewers comments are detailed on the following pages.

            The suggestions/comments of the Reviewers are in black, and our responses are in blue.

Thank you very much for your kind assistance!

Sincerely yours,

Assoc. Prof. Dr. Eng. Simona Popescu

Response to Reviewer 2 Comments

This paper reports the development of Silk fibroin/ZnO coated Ti nanotubes for improvement of the antimicrobial effect of the TI implant. The NT/SF/PD/ZnO coated Ti was characterized surface analysis, electrochemical assessments, and antibacterial assay. The developed NT/SF/PD/ZnO coated Ti was outstanding properties and is potential application for dental implant. The reviewer comments are listed as follows.

Question 1. Biocompatibility

In general, Titanium implant must be biocompatible. In vivo and in vitro experiments using appropriate cells or animals should be needed to verify the novel material's biocompatibility. At least, please discuss the biocompatibility of the material.

 Answer 1: Thank you very much for your suggestions! We agree with the necessity of in vivo and in vitro experiments to verify the biocompatibility of the novel coating. This are planned as the next step, to continue our research. In this first part, we insist on the elaboration methods for the hybrid coating and establishing the principal characteristics that will determine the biological behavior.

We introduced in the Introduction section more details regarding the biocompatibility of the hybrid coatings components.

Furthermore, ZnO is one of the inorganic nanomaterials that the FDA (Food and Drug Administration) has approved for use on the human body.

Silk fibroin (SF), a natural protein spun by silkworms (Bombyx mori) is a biomaterial that has been extensively investigated for biomedical applications. It presents remarkable properties such as non-toxicity, high biocompatibility, adequate biodegradability, superior mechanical and elastic properties and deficient inflammatory reactions.

Therefore, it can be used as a coating material to improve the adhesion and biocompatibility of biomaterial surfaces [36]. PDA coatings are used to bind functional sub-stances (drugs, metal or oxide nanoparticles, hydroxyapatite, etc.) on the surface of metallic biomaterials to improve biocompatibility, osseointegration, cytotoxicity resistance, and antibacterial activity [39,42].

Combining silk fibroin with dopamine, two natural polymers, can result in a material that is biocompatible, environmentally friendly, with enhanced mechanical properties, that can be processed into various forms (films, fibers, gels, sponges) making it suitable for a range of applications, especially in the biomedical field [43].

Question 2. EDX spectra

For surface composition of each sample, only compositional value are listed in Table 1. If possible, please add EDX spectra for each sample. The EDX spectra can clarify the surface composition.

 Answer 2: The EDX spectra for NT/SF-ZnO, NT/ZnO/SF and NT/SF/PD/ZnO have been added in Figure 2 of the manuscript. Thank you very much for your comments!  

Figure 2. SEM images corresponding to a) NT/SF (magnitude of 8236X and 4030X); b) NT/SF-ZnO (magnitude of 7011X and 3829X); c) NT/ZnO/SF (magnitude of 9006X and 3689X); d) NT/SF/PD/ZnO (magnitude of 8806X and 4509X) and EDX diagrams: e) NT/SF-ZnO, f) NT/ZnO/SF and g) NT/SF/PD/ZnO

Question 3. Antimicrobial assessment.

Antimicrobial assay has been performed only at the time 18 hours culture. If possible, please evaluate antimicrobial time dependence.

 Answer 3.  Thank you for your suggestion, we agree that time dependence of antimicrobial effect could bring more information about the mechanism of ZnO action. This paper was more focused on the elaborating, optimizing methods and characterization of the coating and comprises only an antimicrobial test, after first 18 h. This test already provides a first conclusion about the antibacterial behavior of the hybrid coating, because the first hours after the implantation are crucial. We intend to continue this research and we will consider for a future more developed study research that will comprise the evaluation of time dependence antimicrobial effect and also, in vitro test for testing the biocompatibility.

Dr. eng. Simona POPESCU

Associate Professor
The National University of Science and Technology POLITEHNICA

Faculty of Chemical Engineering and Biotechnologies

Department of General Chemistry
1 Polizu Street, Bucharest, 011061
e-mail:  [email protected]   or  [email protected]

Round 2

Reviewer 1 Report

Accept in present form